# Egestion Versus Excretion: A Meta-Analysis Examining Nutrient Release Rates and Ratios across Freshwater Fauna

**Halvor M. Halvorson** [1,2,*] and **Carla L. Atkinson** [3]

1   School of Biological, Environmental, and Earth Sciences, University of Southern Mississippi, 118 College Dr. #5018, Hattiesburg, MS 39402, USA
2   Department of Biology, University of Central Arkansas, 180 Lewis Science Center 201 Donaghey Ave., Conway, AR 72035, USA
3   Department of Biological Sciences, University of Alabama, 2109 Bevill Building, Tuscaloosa, AL 35487, USA; carla.l.atkinson@ua.edu
*   Correspondence: halvorso@gmail.com

**Abstract:** In aquatic settings, animals directly affect ecosystem functions through excretion of dissolved nutrients. However, the comparative role of egestion as an animal-mediated nutrient flux remains understudied. We conducted a literature survey and meta-analysis to directly compare nitrogen (N), phosphorus (P), and N:P of egestion compared to excretion rates and ratios across freshwater animals. Synthesizing 215 datasets across 47 animal species (all primary consumers or omnivores), we show that the total N and P egestion rates exceed inorganic N and P excretion rates but not total N and P excretion rates, and that proportions of P egested compared to excreted depend on body size and animal phylum. We further show that variance of egestion rates is often greater than excretion rates, reflecting greater inter-individual and temporal variation of egestion as a nutrient flux in comparison to excretion. At phylogenetic levels, our analysis suggests that Mollusca exhibit the greatest rates and variance of P egestion relative to excretion, especially compared to Arthropoda. Given quantitative evidence of egestion as a dominant and dynamic animal-mediated nutrient flux, our synthesis demonstrates the need for additional studies of rates, stoichiometry, and roles of animal egestion in aquatic settings.

**Keywords:** consumer-driven nutrient dynamics; stoichiometry; biogeochemistry; rivers/streams; lakes/ponds

## 1. Introduction

Animals can elicit strong, direct effects on nutrient dynamics in aquatic ecosystems by releasing nutrient wastes back into their environments, forming feedbacks on nutrient availability and shaping ecosystem processes [1,2]. Since freshwater ecosystems are often limited by phosphorus (P) and nitrogen (N), rates and ratios of animal nutrient release can be important in determining ecological structure and function [3,4]. Within freshwaters, many studies have quantified roles of animal excretion of dissolved inorganic N (DIN) as ammonium and dissolved inorganic P (DIP) as phosphate, showing direct connections to algal community composition, algal N versus P limitation, and basal resource N and P contents [5,6]. Still other studies have shown a direct contribution of DIN and DIP excretion to food web compartments within riverine ecosystems [7], and ecosystem-level N and P dynamics [8,9]. Despite substantial taxonomic, temporal, and spatial variability that merit further study [10,11], excretion represents a clear pathway for animal community dynamics and evolutionary processes to affect ecosystem processes in many aquatic settings.

Although excretion of DIN and DIP is the best-studied component of consumer-driven nutrient dynamics, animals can also affect nutrient cycling by the production of particulate nutrient wastes such as egesta, exuvia, and carcasses [2]. These primarily organic wastes are less-studied due to a lack of standard quantitative methods, as well as comparatively low bioavailability of organic relative to inorganic N and P, which is assumed to equate to reduced ecological significance (e.g., [12]). However, animal particulate wastes can be highly bioavailable, nutrient-rich, and highly diverse produced via processes including egestion [13] and mortality [14]. For example, rates of animal N and P egestion can equal or exceed rates of DIN and DIP excretion among the few taxa among which the two fluxes have been directly compared [15,16]. Decomposing animal carcasses, too, can supply limiting N and P and elicit enduring effects on recipient ecosystems [17,18]. Many animal particulate wastes are available to microbial heterotrophs and are thus an important nexus between "green" autotrophic and "brown" heterotrophic food webs [19]. Further study of the comparative importance of dissolved versus particulate nutrient wastes will provide a broader understanding of the role of animals in freshwater ecosystems.

Animal egestion contrasts with animal excretion, because the former represents materials which are ingested but not digested/assimilated whereas the latter represents materials which are assimilated but not retained for long-term growth or storage. Given this, relative nutrient fluxes of egesta versus excreta from animals may vary with trophic mode (e.g., herbivore versus predator), body size, and taxonomic identity. Specifically, organisms that face high consumer-resource elemental imbalances (e.g., exhibit high nutrient demands or feed on low-nutrient foods) may release nutrients at lower rates [20,21], but may also release nutrients primarily via egestion, because they are growth- and assimilation-limited, whereas organisms facing lower imbalances may primarily excrete excess nutrients through post-assimilatory regulation [2,22]. Still, animals may partly reduce the effects of imbalances through flexible changes in the gut [23]. To adapt or acclimate to highly imbalanced diets, for example, some animals develop longer digestive tracts, as illustrated by comparisons of herbivorous versus carnivorous fish [24], and among tadpoles reared on low- versus high-N diets [25]. When facing high elemental imbalances, animals may also meet energetic and nutritional needs by increasing feeding rates (compensatory feeding; [26,27]) which combined with low assimilation efficiencies, may result in greater egestion relative to excretion rates [28]. Despite the potential roles of food nutrient content and trophic mode in nutrient release, recent syntheses suggest that excretion rates are only weakly related to trophic imbalances and are best predicted by body size and, to some degree, taxonomy [29,30]. Comparative fluxes of nutrient egestion versus excretion may similarly vary across animals, but to date, no study has examined both fluxes across a diversity of animal taxa ranging in body size and trophic modes.

Here, we conducted a literature survey and meta-analysis of existing studies directly measuring N, P, and N:P egestion and excretion by 47 freshwater animal species. We used our synthesis to directly compare egestion and excretion and test the following predictions: (1) Egestion rates will approximately equal excretion rates across animal taxa, as found in previous comparisons among select taxa [16,17,31]; (2) given the importance of both factors for predicting excretion rates, ratios of nutrient release as egestion, relative to excretion, will be best-predicted by a combination of both body size and taxonomic identity [29,30]; and (3) because egestion is more sensitive to individual-level variation in feeding time and behavior, variance of N and P egestion rates and N:P ratios will be greater than variance of excretion rates and ratios across all taxa.

## 2. Methods

### 2.1. Literature Survey

We conducted a meta-analysis of existing datasets regarding N, P, and N:P excretion and egestion across freshwater animals. We identified publications via the Web of Science database using the search terms TS = ((egest* OR excret* OR defecat* OR feces OR faeces OR recycl*) AND (stoichiometr* OR

nitrogen OR phosphorus OR nutrient) AND (lake OR stream OR wetland OR freshwater)). This search was conducted on 23 February 2019 and resulted in 2192 potential publications. We also supplemented this collection with a search using Google Scholar, which identified 575 potential publications including theses/dissertations in a search on 27 February 2019. We scanned titles and abstracts to determine inclusion of all publications. To fit criteria for meta-analysis, studies must have been conducted using freshwater animals and must have directly measured both excretion and egestion as N or P-specific release rates or N:P ratios, either in the laboratory or the field, from the same study population (often both terms were measured simultaneously from the same individuals). We included both field and laboratory trials because of the scarcity of existing field egestion data from many taxa. However, our meta-analysis was not exhaustive because our use of habitat-specific search terms excluded some laboratory data. Notably, our analysis did not include any fish aquaculture studies which have measured excretion and egestion rates (e.g., [32,33]), but all studies in our meta-analysis were conducted using organisms collected originally from the field.

A total of 21 peer-reviewed publications or theses/dissertations fit criteria for inclusion in the meta-analysis, which we supplemented with three unpublished datasets of our own (Table 1). From each publication, we recorded study animal(s) species identity, individual dry mass, study site and collection date, whether trials were conducted in the laboratory or in the field, sample sizes, temperature during nutrient release trials, N, P, and N:P excretion and egestion rates or ratios, and body %N, %P, and N:P as well as diet %N, %P, and N:P and identity. All studies reported total N or P egestion rates and did not separate by inorganic versus organic egested material. Most studies reported excretion rates exclusively of DIN or DIP (primarily $N-NH_4$, but sometimes $N-[NH_4 + NO_3]$; and $P-PO_4$) whereas others reported excretion based on total N and P (TN and TP). We collected data regarding both forms of excretion and separately analyzed datasets from each form (see below). We contacted corresponding authors as necessary to obtain raw data from publications. We then calculated the mean ($\mu$), standard deviation (SD), and coefficient of variation (CV) of nutrient release rates and ratios from each study population. We defined a "study population" as the focal unit for our meta-analysis, identified as a distinct species population studied at a given site on a given sampling event, or reared under distinct conditions (e.g., temperature or diet) in the laboratory. In total, our synthesis included 215 study populations from 47 different animal species.

**Table 1.** Summary of datasets included in the meta-analysis of egestion and excretion rates across freshwater animal taxa. See Supplementary Material 1 for a comprehensive summary of release ratios and relative variance across all study populations.

| Source Study | Setting | Location | Phyla | # Species | # Study Populations |
|---|---|---|---|---|---|
| Andre et al. 2003 [34] | Field | Lake Malawi, Africa | Chordata | 5 | 5 |
| Atkinson et al. 2018 [35]; Atkinson et al. unpub. | Field | Oklahoma, USA | Mollusca | 6 | 15 |
| Atkinson et al. unpub. | Field | Alabama, USA | Mollusca | 10 | 28 |
| Christian 2002 [36]; Christian et al. 2008 [37] | Field | Arkansas and Ohio, USA | Mollusca | 4 | 27 |
| Cyr et al. 2017 [38] | Field | New Zealand | Mollusca | 1 | 5 |
| Greene 2015 [39] | Field | Arizona, USA | Chordata | 2 | 2 |
| Hall et al. 2003 [40] | Field | Wyoming, USA | Mollusca | 1 | 1 |
| Halvorson et al. 2015 [16] | Lab | Arkansas, USA | Arthropoda | 2 | 14 |
| Halvorson et al. 2017 [41] | Lab | Arkansas, USA | Arthropoda | 1 | 3 |
| Halvorson et al. unpub. | Field | Arkansas, USA | Arthropoda | 7 | 10 |
| Halvorson et al. unpub. | Lab | Arkansas, USA | Arthropoda | 2 | 8 |
| Hoellein et al. 2017 [42] | Field | Illinois, USA | Mollusca | 2 | 2 |
| Hood et al. 2014 [15] | Lab and Field | California and Minnesota, USA | Arthropoda | 3 | 21 |
| Liess 2014 [31] | Lab | Sweden | Mollusca | 2 | 2 |
| Liess et al. 2015 [25] | Lab | Sweden | Chordata | 1 | 8 |
| Mas-Marti et al. 2015 [43] | Lab | Spain | Arthropoda | 1 | 4 |
| McLeay et al. 2019 [44]; McLeay 2017 [45] | Field | Georgia, USA | Chordata | 1 | 17 |
| Mosley and Bootsma 2015 [46] | Field | Lake Michigan, USA | Mollusca | 1 | 15 |
| Norlin et al. 2016 [47] | Field | Sweden | Chordata | 1 | 4 |
| Ozersky et al. 2015 [48] | Field | Ontario, Canada | Mollusca | 2 | 6 |
| Subalusky et al. 2015 [49] | Lab | Milwaukee Zoo, USA | Chordata | 1 | 1 |
| Vanderploeg et al. 2017 [50] | Field | Lake Erie, USA | Mollusca | 1 | 1 |
| Villanueva et al. 2011 [51] | Lab | Portugal | Arthropoda | 1 | 6 |
| Williamson and Ozersky in press [52]; Williamson 2017 [53] | Field | Minnesota, USA | Mollusca | 1 | 10 |

## 2.2. Calculations

For each study population, we determined a release ratio as the direct comparison between paired N or P egestion and excretion rates or N:P ratios. Release ratios were calculated using log response ratios:

$$Release\ ratio = \ln\frac{\mu_{egest}}{\mu_{excrete}} \qquad (1)$$

where $\mu$ indicates the mean rate or ratio of elements released as egesta (numerator) or excreta (denominator) within a given study population. In this way, values above zero indicate a rate or ratio of egestion exceeding excretion.

We also calculated variances of release ratios using the equation

$$Variance = \frac{SD^2_{egest}}{n_{egest}\mu^2_{egest}} + \frac{SD^2_{excrete}}{n_{excrete}\mu^2_{excrete}} \qquad (2)$$

where *SD* indicates standard deviation of egestion or excretion rates or ratios within the study population, *n* indicates sample size (number of nutrient release trials), and μ indicates the study population mean.

To quantify the relative variance of egestion compared to excretion rates and ratios, we also calculated the relative variance of each release pathway for each study population using Equation (3)

$$Release\ relative\ variance = ln\frac{CV_{egest}}{CV_{excrete}} \qquad (3)$$

where *CV* indicates the coefficient of variation (*SD* divided by $\mu$) of the mean rate or ratio of elements released as egesta (numerator) or excreta (denominator) within a given study population. Thus, positive values of relative variance indicate egestion rates are more variable than excretion rates or ratios.

Finally, as an exploratory analysis, we quantified consumer-resource elemental imbalance for all study populations that included data regarding consumer and resource %N, %P, or N:P. We calculated elemental imbalance using Equation (4)

$$Elemental\ imbalance = ln\frac{X_{consumer}}{X_{resource}} \qquad (4)$$

where *X* indicates %N, %P or N:P of consumers and resources [54,55]. Elemental imbalances should thus be positively related to differences in elemental contents of consumers relative to their resources. We note this calculation assumes that the identified nutrient is limiting and thus does not account for stronger limitation by other elements, such as when both N and P are highly imbalanced.

## 2.3. Statistical Analyses

We used weighted mixed effects models to test differences of N, P, and N:P release ratios and relative variances across species, comparing four candidate models using the function lmer in the R package lme4 [56]. Model 1 (null model) included taxonomic family and source study as separate random effects terms, but included no fixed effects. Three other models were compared directly to Model 1; these models included random effects of family and source study, but additionally included fixed effects of phylum (Model 2; three phyla: Mollusca, Chordata, or Arthropoda), $\log_{10}$ body dry mass (Model 3) or both phylum and $\log_{10}$ body dry mass as additive effects (Model 4). We determined the best-fit model based on Akaike Information Criteria (AIC) [57]. Models 2–4 were compared to Model 1 using a chi-square test to determine whether the fixed effect(s) provided a significant fit to the data, relative to their absence. We also determined conditional $R^2$ of each model using the R package MuMIn [58]. We weighted all models by the inverse of variance (release ratio models) or by sample size (relative variance models).

Due to a smaller and less diverse number of datasets measuring excretion as TN or TP, we only fit the null model (Model 1) to N, P, and N:P release ratios and relative variance terms based on TN and TP excretion rates. Within sets of models based on smaller datasets, we avoided singular model fits by removing source study as a random effects variable from all models. After fitting all mixed effects models, we used Model 1 or the better-fit model, where appropriate, to determine the model intercept. We subsequently used two-tailed *t*-tests ($\alpha = 0.05$) to compare model intercepts to zero, indicating a null hypothesis of release ratios or relative variance equal to zero, signifying no difference in mean or variance of N, P, or N:P egestion relative to excretion across all species. Where mixed effects modeling indicated a significant effect of phylum on release ratios or relative variance, we used Tukey's honestly significant difference (HSD) to identify phylum-level differences.

As a supplement to the above mixed effects models, we also produced scatterplots and conducted Pearson's correlation tests to examine relationships between consumer-resource elemental imbalance and release ratios. A summary of all datasets used in the meta-analysis may be found in Table 1 and Supplementary Material 1. All statistical analyses were conducted in R version 3.5.1 [59].

## 3. Results

### 3.1. Literature Survey

The literature survey and data collation resulted in paired means of N, P, or N:P egestion and excretion across 215 study populations of 47 freshwater animal species representing three phyla (Chordata, Mollusca, and Arthropoda), ranging widely in individual body dry mass from 0.448 mg (*Allocapnia* spp.) to 529.1 kg (*Hippopotamus amphibus*). All taxa were primary consumers or omnivores, with no carnivores included. The majority of datasets came from the field, as evidenced by 16 studies conducted in the field and 159 out of 215 (74%) datasets collected from field individuals (Table 1). The majority of datasets reported excretion rates as DIN, DIP, or DIN:DIP (62%–77% of all datasets) compared to TN, TP, or TN:TP (21%–28% of all datasets). Across all species, rates of N and P egestion and excretion scaled positively with body mass (Figure S1 Supplementary Material 2).

### 3.2. Nutrient Release Ratios

Across species, model comparisons indicated that N release ratios based on DIN excretion were best predicted by Model 1 (null model), although Model 3 was within 2 AIC of Model 1, suggesting a possible role of body size but not phylum in N release patterns (Table 2). The mean ± SE global intercept for Model 1 was 0.782 ± 0.271, a ratio significantly greater than a null hypothesis of zero ($t_{1,126} = 2.89$; $p = 0.005$; Figure 1a, Table S1). Thus, across all taxa, N egestion rates exceeded DIN excretion rates. By contrast, N release ratios based on TN excretion were on average negative (mean ± SE = −0.558 ± 0.319) and did not differ from zero ($p = 0.092$; Figure 1a, Table S1).

**Table 2.** Summary of weighted mixed effects models predicting nitrogen (N), phosphorus (P), and N:P release ratios of egestion compared dissolved inorganic N (DIN), dissolved inorganic P (DIP), and DIN:DIP excretion across animals. Boldface indicates best-fit models based on lowest Akaike Information Criteria (AIC) scores. See Table S1 for model intercepts.

| Response | Model Structure | AIC | Chi-Square | *p*-Value | R-Squared |
|---|---|---|---|---|---|
| N release ratio | **1\|Source Study + 1\|Family** | **591.2** | | | **0.03** |
| | Log Dry Mass + 1\|Source Study + 1\|Family | 592.4 | 0.82 | 0.365 | 0.03 |
| | Phylum + 1\|Source Study + 1\|Family | 594.6 | 0.00 | 1.000 | 0.03 |
| | Log Dry Mass + Phylum + 1\|Source Study + 1\|Family | 595.7 | 0.85 | 0.356 | 0.03 |
| P release ratio | 1\|Source Study + 1\|Family | 1672.0 | | | <0.01 |
| | Log Dry Mass + 1\|Source Study + 1\|Family | 549.6 | 1124.4 | <0.001 | 0.24 |
| | Phylum + 1\|Source Study + 1\|Family | 1668.2 | 0.00 | 1.000 | <0.01 |
| | **Log Dry Mass + Phylum + 1\|Source Study + 1\|Family** | **546.9** | **1123.3** | **<0.001** | **0.22** |
| | **1\|Source Study + 1\|Family** | **425.1** | | | **0.06** |
| N:P release ratio | Log Dry Mass + 1\|Source Study + 1\|Family | 428.4 | 0.00 | 1.000 | 0.06 |
| | Phylum + 1\|Source Study + 1\|Family | 428.1 | 2.3 | 0.131 | 0.07 |
| | Log Dry Mass + Phylum + 1\|Source Study + 1\|Family | 431.2 | 0.00 | 1.000 | 0.07 |

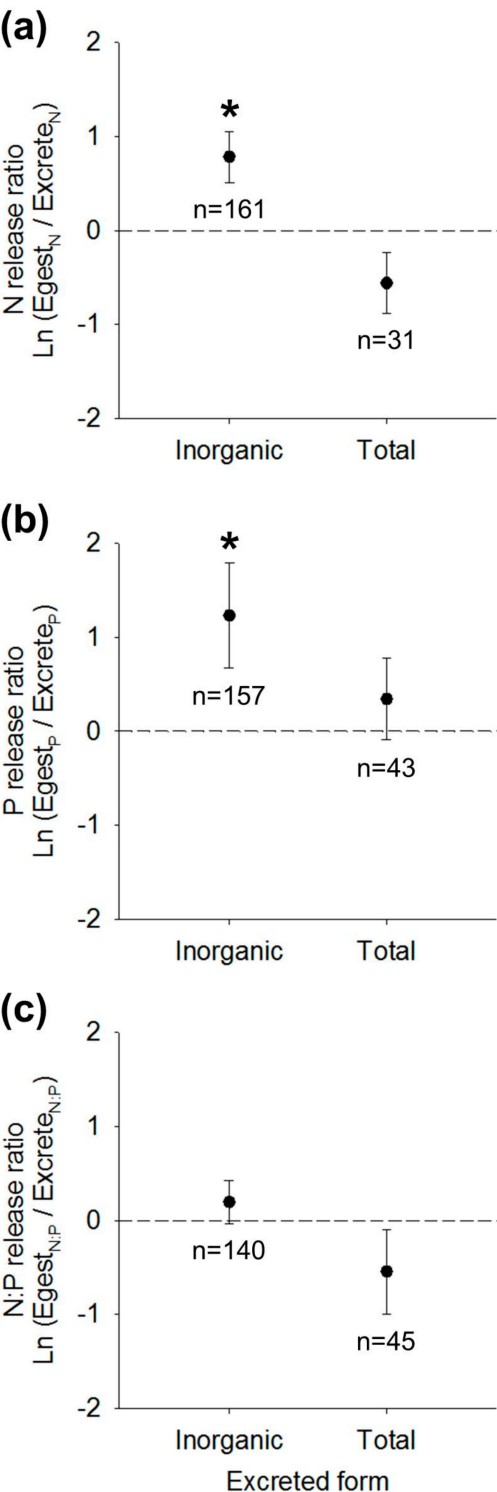

**Figure 1.** Mean ± SE global intercepts of best-fit weighted mixed effects models predicting (**a**) N release ratios, (**b**) P release ratios, or (**c**) N:P release ratios of egestion compared to excretion. In each panel, means describe release ratios calculated from total nutrient egestion rates relative to (left) strictly inorganic nutrient excretion or (right) total nutrient excretion. Positive values indicate egestion rates or ratios exceed excretion rates or ratios. Sample sizes *n* (number of study populations) are designated for each mean and asterisks designate mean intercepts significantly different from zero (two-tailed *t*-test; $p < 0.05$). See Table 2 and Table S1 for associated model fits.

The best-fit model predicting P release ratios based on DIP excretion was Model 4, a model including additive terms of log body mass and phylum (Table 2). Phosphorus release ratios were negatively related to $\log_{10}$ individual body dry mass in mg (slope = −1.083 ± 0.002) and Mollusca exhibited higher P release ratios than Chordata, followed by Arthropoda, but P release ratios did not differ significantly across phyla (Tukey's HSD; $p > 0.05$; Figure 2a). Across all taxa, Model 4 predicted P release ratios of 1.23 ± 0.56, a ratio significantly greater than zero ($t_{1,122} = 2.21$; $p = 0.029$; Figure 1b, Table S1). Thus, similar to N release ratios, rates of P egestion exceeded rates of DIP excretion. Based on TP excretion rates, P release ratios were net positive (0.343 ± 0.431) and did not differ from zero ($p = 0.430$; Figure 1b, Table S1).

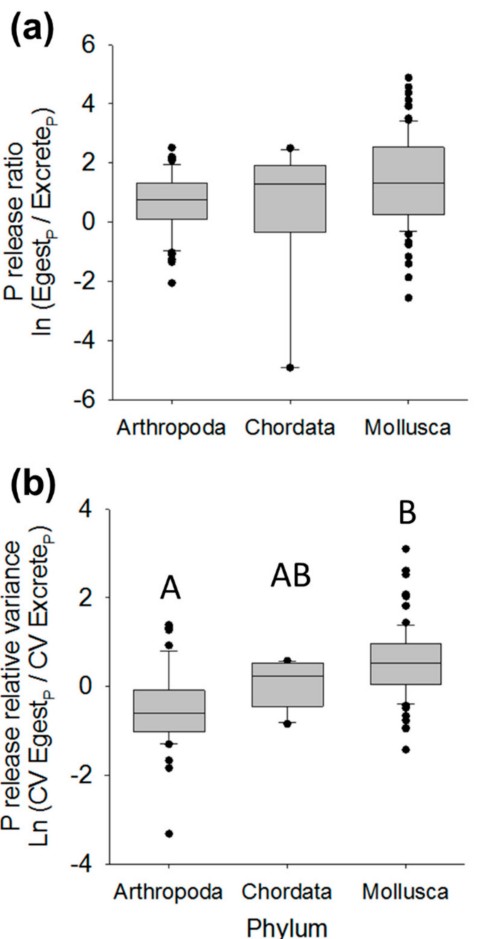

**Figure 2.** Box and whisker plots of P release ratios (**a**) or P relative release variance (**b**) based on total P egestion relative to inorganic P excretion across distinct phyla. In both (**a**) and (**b**), weighted mixed effects models including both phylum and body size provided the best-fit explanation of release ratios or relative variance (Tables 2 and 3). Upper-case letters A and B in panel (**b**) indicate significant differences across phyla based on Tukey's honestly significant difference (HSD) ($p < 0.05$), with Arthropoda exhibiting significantly lower P release relative variance compared to Mollusca, and Chordata statistically similar to both groups. There are no upper-case letters in panel (**a**) because Phyla did not significantly differ in P release ratios.

Release ratios of N:P egestion relative to DIN:DIP excretion were best-explained by Model 1, indicating little variance attributable to phylum or body size (Table 2). Based on Model 1, N:P release ratios across all taxa were on average positive (0.195 ± 0.230) but did not differ from zero ($t_{1,108} = 0.85$, $p = 0.397$; Table S1, Figure 1c). Release ratios based on TN:TP excretion shifted to negative (−0.542 ±

0.450) and did not differ from zero ($t_{1,40} = -1.21$, $p = 0.233$; Table S1, Figure 1c). Thus, N:P egestion did not differ from N:P excretion as either inorganic or total forms of excretion.

Among the subset of taxa for which imbalances could be calculated, patterns in N, P, and N:P release ratios varied widely across taxa, but covaried weakly with N, P, and N:P consumer-resource imbalance despite a wide range of imbalances (Figure S2). Notably, variance in P release ratios increased with greater P imbalances, evident by high variance among Mollusca compared to Chordata (Figure S2b). Release ratios of N:P were weakly negatively related to N:P imbalance (Figure S2c).

*3.3. Nutrient Release Relative Variance*

As indicators of the comparative variance of egestion relative to excretion, relative variance of N release based on DIN excretion was best-predicted by Model 1, but Model 3 was also within 2 AIC, suggesting a potential role of body size (Table 3). Within Model 1, N release relative variance was 0.236 ± 0.125, which was significantly greater than zero ($t_{1,137} = 2.60$, $p = 0.010$; Figure 3a, Table S2), indicating greater variance of N egestion than DIN excretion rates across all taxa. Based on TN excretion rates, N release relative variance was also positive (0.368 ± 0.326) but did not differ from zero ($t_{1,24} = 1.13$, $p = 0.270$; Figure 3a, Table S2).

Similar to P release ratios, relative variance of P release based on DIP excretion was best explained by Model 4, which included effects of both phylum and body mass (Table 3). Within 2 AIC was also Model 3, indicating consistent support for an effect of body mass on P release variance (Table 3). Model 4 indicated a negative slope effect of $\log_{10}$ body mass on relative variance of P release (slope = −0.403 ± 0.126) and Tukey's HSD indicated significantly greater relative variance of P release among Mollusca compared to Arthropoda (Figure 2b). The relative variance of P release was on average positive (0.417 ± 0.229), but did not differ significantly from zero ($t_{1,122} = 1.82$; $p = 0.071$), indicating no difference in variance of P egestion relative to DIP excretion across all taxa (Figure 3b, Table S2). Relative variance of P release as TP excretion also was positive (0.176 ± 0.157) but, like for DIP excretion, did not differ from zero ($t_{1,26} = 1.1$, $p = 0.273$; Figure 3b, Table S2).

Relative variance of N:P egestion versus DIN:DIP excretion was also best-predicted by Model 1 (Table 3) and relative variance was negative (–0.189 ± 0.210) but not significantly different from zero ($t_{1,87} = -0.9$, $p = 0.372$). Similarly, relative variance of N:P egestion and TN:TP excretion was negative (–0.187 ± 0.473) and did not differ from zero ($t_{1,34} = -0.40$, $p = 0.695$; Figure 3c, Table S2).

**Table 3.** Summary of weighted mixed effects models used to predict N, P, and N:P release relative variance of egestion compared to excretion across animal taxa. Best-fit models based on lowest AIC scores are highlighted in bold. Only models predicting relative variance based on inorganic nutrient excretion were compared. See Table S2 for best-fit model global intercepts.

| Response | Model Structure | AIC | Chi-Square | *p*-Value | R-Squared |
|---|---|---|---|---|---|
| N release relative variance | **1\|Family** | **348.9** | | | **0.03** |
| | Log Dry Mass + 1\|Family | 350.5 | 0.5 | 0.496 | 0.03 |
| | Phylum + 1\|Family | 354.9 | 0.0 | 1.000 | 0.03 |
| | Log Dry Mass + Phylum + 1\|Family | 357.5 | 0.0 | 1.000 | 0.04 |
| P release relative variance | 1\|Family | 337.0 | | | 0.06 |
| | Log Dry Mass + 1\|Family | 334.8 | 4.2 | 0.040 | 0.12 |
| | Phylum + 1\|Family | 338.8 | 0.0 | 1.000 | 0.06 |
| | **Log Dry Mass + Phylum + 1\|Family** | **333.7** | **7.1** | **0.008** | **0.10** |
| N:P release relative variance | **1\|Family** | **255.6** | | | **0.05** |
| | Log Dry Mass + 1\|Family | 259.4 | 0.0 | 1.000 | 0.05 |
| | Phylum + 1\|Family | 258.9 | 2.5 | 0.117 | 0.05 |
| | Log Dry Mass + Phylum + 1\|Family | 262.6 | 0.0 | 1.000 | 0.05 |

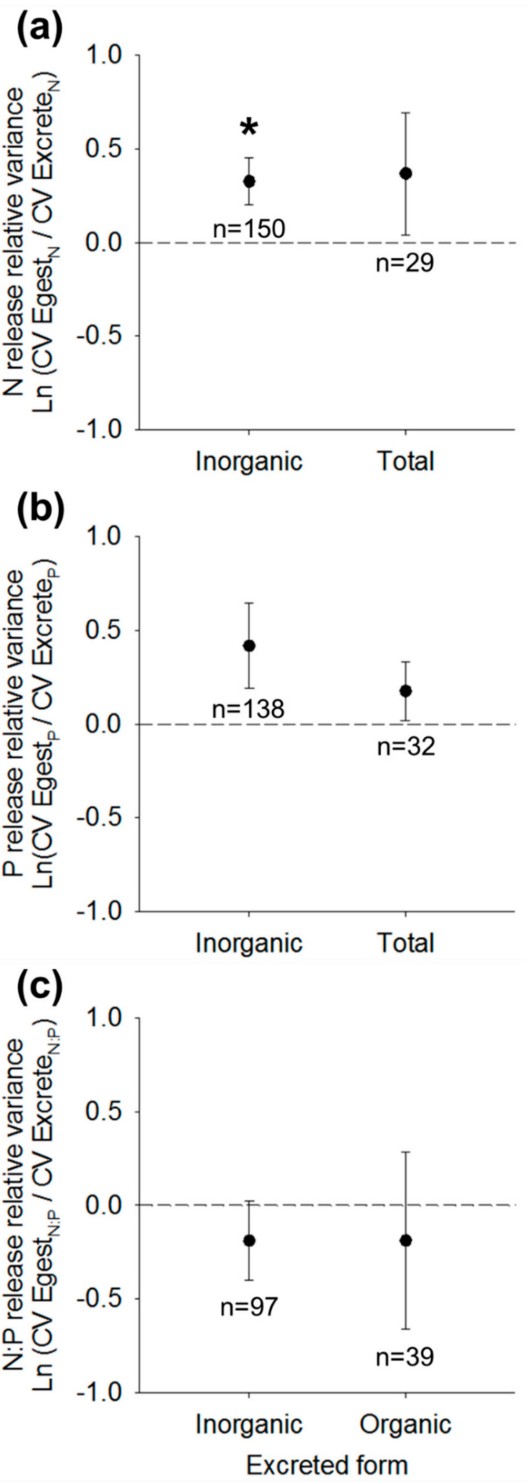

**Figure 3.** Mean ± SE global intercepts of best-fit weighted mixed effects models predicting (**a**) N relative release variance, (**b**) P relative release variance, or (**c**) N:P relative release variance of egestion compared to excretion. In each panel, means describe relative variance calculated from variance of total nutrient egestion relative to (left) strictly inorganic nutrient excretion or (right) total nutrient excretion. Positive values indicate variance of egestion rates or ratios exceed variance of excretion rates or ratios. Sample sizes *n* (number of study populations) are designated for each mean and asterisks designate mean intercepts significantly different from zero (two-tailed *t*-test; $p < 0.05$). See Table 3 and Table S2 for associated model fits.

## 4. Discussion

We found that across multiple freshwater animal species, particulate N and P fluxes in the form of egesta exceed DIN and DIP excretion fluxes. Due to a general lack of data, it is commonly assumed that excretion is the dominant nutrient release flux by aquatic animals. Indeed, many studies have empirically shown excretion to be important at ecosystem levels [7,10,60]. However, recent empirical studies also indicate egestion and excretion rates are approximately equal among many taxa, suggesting that egestion may be an important overlooked pathway of nutrient cycling in aquatic settings [15,16,31]. Drawn from a wide array of predominately primary consumer taxa, our results show that egestion is a major flux measured from animals in freshwater settings. Among a smaller number of datasets measuring TN or TP excretion, we also show that total nutrient egestion rates are similar to total excretion rates, affirming dissolved organic N and P excretion as an important release flux across freshwater animals [61]. Furthermore, our synthesis shows that the variance of N egestion exceeds variance of DIN excretion rates, and we reveal major roles of body size and phylogeny in both relative rates and variance of P egestion compared to DIP excretion. Given this quantitative evidence of egestion as an important and dynamic animal-mediated nutrient flux, our study affirms the need for additional studies of animal egestion in a context additional to animal excretion of dissolved nutrients in aquatic ecosystems [2,13].

While the contributions of egesta to ecosystem nutrient fluxes remain poorly studied in comparison to excretion, it is generally assumed that egesta are not as important, as egesta are often considered low-N and P, recalcitrant, and not as bioavailable as dissolved excreta [2,12]. We show that, among species which have been measured for rates of both egestion and inorganic nutrient excretion, egestion rates often exceed excretion rates. At the physiological level, the quantitative importance of egestion reflects several patterns. First, many organisms are likely unable to completely digest and assimilate ingested N and P, owing to the inefficiency of digestive enzymes and uptake pathways within the gut, combined with selective investment in uptake of limiting nutrients, which can cause non-limiting nutrients to be egested instead of assimilated [23,62]. Second, many organisms may have evolved to undergo faster (e.g., compensatory) feeding instead of maximizing assimilation efficiency, causing the majority of ingested nutrients to pass through the gut [26–28]. Finally, all of the animals included in our study were primary herbivores or detritivores with the datasets not including any higher-level consumers. Among carnivores, egestion rates may be lower than excretion rates due to greater assimilation efficiency of nutrient-rich animal tissues [63], but N:P ratios of egestion may be lower than N:P of excretion among some carnivores because P-rich bone is difficult to assimilate.

At the ecosystem level, our results also establish that egestion may be equally or more important than excretion within the nutrient budgets of aquatic ecosystems—yet, the two pathways of nutrient release may exhibit contrasting ecological implications. Broadly, DIN and DIP in excreta should be more bioavailable than N or P in egesta, because the latter is mostly organic and requires breakdown by heterotrophs [12]. However, this contrast may depend on the proportions of egested N and P that are inorganic versus organic, which may depend on trophic level and remains poorly understood. Previous work suggests that egesta can accrue in depositional zones and may act as sinks rather than sources of dissolved inorganic nutrients, serving as long-term stores of organic nutrients [13,64]. These differing effects of excretion versus egestion are key to understanding benthic-pelagic coupling, such as the role of sessile filter-feeders in spurring sediment biogeochemical processes [48,65,66]. However, Halvorson et al. (2017) [13] noted that whether egesta became a sink for inorganic nutrients, and whether egested nutrients are mineralized, varies as a result of the taxonomic identity of the source animal. Further, few studies have considered egesta as a potential resource for microbes or other animals [67,68]. Our study calls for further study of the diverse fates of animal egesta in aquatic ecosystems, to better link animal physiological processes with ecosystem functions such as nutrient cycling and the interaction between autotrophic versus heterotrophic ecosystem processes [19,61].

As shown in previous work on excretion rates across broad sets of freshwater taxa and marine invertebrates and fish [29,30], our study shows that body size and taxonomic identity play a role in

determining nutrient fluxes by animals. In particular, P egesta fluxes relative to excreta were best predicted by both taxonomy and body size, exhibiting greater release ratios and relative variance among small-bodied taxa. The negative effect of body size on release ratios suggests that P excretion and egestion rates may scale differently with body size—generally, smaller-bodied species exhibit proportionally greater P egestion compared to larger-bodied species. These patterns may reflect contrasting scaling of gut length versus metabolic rates with increasing body size [69]. For example, small-bodied taxa may exhibit greater post-assimilatory demands of P to support faster growth of P-rich tissues, thus reducing P excretion rates [70–72]. Alternatively, body size effects may reflect trophic differences between small taxa that maximize feeding rates at the expense of assimilation, versus larger taxa that have evolved to enhance assimilation efficiencies, e.g., by increasing gut length [24]. While both egestion and excretion release scale approximately similarly with body size when all taxa are pooled together (Figure S1), body size effects in our analysis are significant after accounting for different random intercepts fit to each taxonomic family, because our models treated taxonomic family as a random effect.

We further observed phylogenetic differences wherein Mollusca exhibit greater rates of P egestion, relative to DIP excretion, compared to Chordata and especially compared to Arthropoda. At a methodological level, the Mollusca datasets may have resulted in proportionally greater P egestion due to the inclusion of both pseudofeces and biodeposits in egestion measures from Bivalves. Molluscan pseudofeces represent material filtered but subsequently expelled in mucus, before true ingestion, and thus do not represent truly egested material but contain measurable amounts of N and P [42]. However, the lack of phylum differences with respect to N or N:P release suggest additional phylogenetic drivers specific to P release, perhaps tied to phylum differences in body size, diet, or growth rates that are coupled to organism P demands, but not to N or N:P demands [70,73]. Still, our study suggests a minimal role of diet, because elemental imbalances were not related to release ratios across taxa for which we were able to calculate imbalances.

Our meta-analysis further tested the prediction that egestion rates and ratios would be more variable than excretion rates and ratios across all animal species. We found that release relative variances were most often positive, supporting our prediction and indicating greater within-population variability of egestion compared to excretion. This suggests that, within species or study populations, excretion may be a relatively more uniform pathway of nutrient release compared to egestion. Greater variation of egestion may reflect higher temporal variation and individual-level differences in consumption rates and assimilation efficiency, body size, and feeding mode that are diminished post-assimilation due to homeostatic regulation which is comparatively uniform and temporally stable across individuals. While intra-specific and -population variation of stoichiometric traits remain understudied compared to inter-specific variation [74–76], greater individual-level variation of egestion may promote comparatively greater spatial and temporal heterogeneity of community and ecosystem processes affected by egestion [77]. At a methodological level, studies should investigate why egestion rates exhibit high variation, because some degree of variation may be due to a lack of standard methods for measuring egestion rates across taxa, in comparison to excretion for which methods are well-developed and standardized [78,79]. Interestingly, we found no difference in variation of egestion versus excretion N:P ratios, suggesting that methodological constraints and/or inter-individual variation may affect the rate but not the stoichiometry of release. Indeed, our study shows that variation of P egestion rates is greater among smaller-bodied taxa and among Mollusca compared to Arthropoda, which may reflect either the greater proportion of these datasets derived from the field compared to the laboratory (Table 1), or greater temporal and individual variation in feeding and assimilation processes among small-bodied individuals and Molluscs.

Our study provides the first systematic comparison of egestion and excretion rates and ratios across a diversity of animal taxa and, as such, provides several directions for future research in consumer-driven nutrient dynamics. First, our literature survey revealed many studies reporting excretion data, but was limited by a lack of paired egestion data from many taxa. This limitation will only be addressed by

methods development and measures of egestion simultaneous to excretion, with field-collected data needed wherever methods allow [30]. Future studies must also broaden the diversity of taxa considered to major taxonomic groups that remain poorly represented within our dataset—large-bodied Chordates including reptiles, mammals, and fish, many Arthropod groups (crayfish, stoneflies, beetles, mayflies), Mollusca (snails), and Annelids need better representation. Our study did not include some existing data from fish drawn from aquaculture studies. Finally, the quantitative importance of egestion compared to excretion highlights the need for further ecosystem-level studies of the ecological roles of egestion. These roles are poorly understood and merit continued study of transport/deposition processes, food web significance, and nutrient turnover [13,67]. Studies of egestion will increase scientific understanding of animals' diverse roles in ecosystems, expanding from a historic emphasis on dissolved nutrients to particulate wastes that are linked with animal phylogeny, body size, and other traits and are potentially important within many ecosystem processes [2,80,81].

**Supplementary Materials:** The following are available online at http://www.mdpi.com/1424-2818/11/10/189/s1, Figure S1: Scatterplots of mean log-transformed N and P excretion and egestion rates and individual body dry mass across all study populations included in the meta-analysis. Panels are arranged into (a) N egestion and dissolved inorganic N (DIN) excretion, (b) N egestion and total N (TN) excretion, (c) P egestion and dissolved inorganic P (DIP) excretion, and (d) P egestion and total P (TP) excretion. Equations next to legend identifications describe linear regression statistics for each pathway of nutrient release as a function of body mass. Only populations from which both excretion and egestion were measured are included, Figure S2: Scatterplots of (a) N release ratios, (b) P release ratios, or (c) N:P release ratios of egestion relative to excretion across degrees of consumer-resource elemental imbalance. Pearson's correlation coefficients and *p*-values are designated for each dataset (all phyla included) within each panel, Table S1: Summary of best-fit mixed effects model intercepts for all analyses of release ratios, Table S2: Summary of best-fit mixed effects model intercepts for all analyses of release relative variance.

**Author Contributions:** Conceptualization, H.M.H. and C.L.A.; Methodology, H.M.H. and C.L.A.; Investigation, H.M.H. and C.L.A.; Formal Analysis, H.M.H.; Writing, H.M.H. and C.L.A.

**Funding:** This research was funded by the U.S. NATIONAL SCIENCE FOUNDATION DEB #1501703 and DEB #1457217 support of HMH and the U.S. NATIONAL SCIENCE FOUNDATION DEB #1831512 support of CLA. Publication costs were supported by the ALABAMA WATER INSTITUTE at the University of Alabama.

**Acknowledgments:** Thank you to Rachel Moore, Grant White, and Brian Van Ee for assistance collecting unpublished data, as well as all authors who shared raw data or provided input including Tim Hoellein, Jim Hood, Antonia Liess, and Ted Ozersky.

**Conflicts of Interest:** The authors declare no conflict of interest.

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
