# Peer review of "Egestion Versus Excretion: A Meta-Analysis Examining Nutrient Release Rates and Ratios across Freshwater Fauna"

_diversity, doi:10.3390/d11100189_

Round 1
Reviewer 1 Report
I was quite excited to see this paper as I have been thinking that a study of this sort has been needed for many years now. Indeed, I think a direct comparison of excretion and egestion rates and ratios will be a valuable contribution to the field and help us think about many aspects of stoichiometric models and nutrient budgets of ecosystems.
Much of the author’s thoughts about this problem and results are similar to what I had expected based on what little is known about this subject and what theory would predict, but compiling it together in a formal meta-analysis will help cement some small pieces together to move this area of research forward. I found the paper overall to be well-written and most ideas were quite clear. A few exceptions are noted below in my comments.
My primary concern with the paper is that it focuses on primary consumers but this focus and its implications for the results are not stated clearly enough. I personally don’t think that the results for species at higher trophic levels would be the same, and it’s not clear why the authors did exclude such studies. I was able to find a number of studies that would seem to fit the inclusion criteria (see examples listed below), and my only guess is that they did not show up within the author’s search terms because of the habitat-specific keywords. If the authors understandably do not wish to redo their search and analysis, I would then suggest focusing more clearly on primary consumers, perhaps even only on the field studies which I am more confident were searched adequately, and discussing the implications of this focus more explicitly.
Abstract
The second sentence (and the rest of the abstract) didn’t clearly follow the first in my mind. I agree that the importance of egestion is not nearly as well-known in aquatic ecosystems, but this study does not address that question. On the other hand, it’s equally true that we (aquatic ecologists) don’t consider egestion rates as much as we think about egestion rates, and that is a better lead in to the gap this paper fills.
Introduction
The second paragraph seems to conflate organic waste and egestion. As I understand it, all egestion is not organic and all organic waste is not egested. More care should be taken here to set up clearly how these terms will be defined and used throughout the manuscript.
Line 48-49: Here this is particularly confusing. Are you talking about organic forms of N and P being released by carcasses, or organic matter in general? The Kohler et al. reference only appears to include data for inorganic forms of N and P, which doesn’t seem consistent with the way you are calling them organic wastes. Maybe this group of waste products should be referred to as “particulate animal wastes” to distinguish them from excretion or it needs to be more clearly described how they represent only organic wastes.
Line 74: Your first hypothesis seems more like an assumption that is made by people who don’t measure both rates. If there is some rationale to support this hypothesis it would help to present it here.
Line 76: I have a hard time following the third hypothesis. This would make sense if you are talking about the imbalance for the limiting nutrient and growth gross efficiency or overall dietary retention, but it sounds like you are saying that the assimilation efficiency of a particular nutrient will decrease when the imbalance is higher? To me it seems like the direction of this relationship would depend highly on which nutrient is limiting growth, so it seems unlikely that you’d find a clear pattern here.
Methods
Your search appears to have missed many papers that do present both rates in fishes in laboratory settings. This could be an artifact of your use of the habitat terms in your keywords but then deciding to include laboratory studies. Your laboratory studies were all done by ecologists who work on these questions, but they are also well-studied by physiologists and aquaculturists in lab settings. Check out these examples I was able to find with similar search terms without the freshwater habitat keywords:
Bureau, D. P., & Cho, C. Y. (1999). Phosphorus utilization by rainbow trout (Oncorhynchus mykiss): estimation of dissolved phosphorus waste output. Aquaculture, 179(1-4), 127-140.
Green, J. A., Hardy, R. W., & Brannon, E. L. (2002). Effects of dietary phosphorus and lipid levels on utilization and excretion of phosphorus and nitrogen by rainbow trout (Oncorhynchus mykiss). 1. Laboratory‐scale study. Aquaculture nutrition, 8(4), 279-290.
Médale, F., Boujard, T., Vallée, F., Blanc, D., Mambrini, M., Roem, A., & Kaushik, S. J. (1998). Voluntary feed intake, nitrogen and phosphorus losses in rainbow trout (Oncorhynchus mykiss) fed increasing dietary levels of soy protein concentrate. Aquatic Living Resources, 11(4), 239-246.
Ogunkoya, A. E., Page, G. I., Adewolu, M. A., & Bureau, D. P. (2006). Dietary incorporation of soybean meal and exogenous enzyme cocktail can affect physical characteristics of faecal material egested by rainbow trout (Oncorhynchus mykiss). Aquaculture, 254(1-4), 466-475.
Schneider, O., Amirkolaie, A. K., Vera‐Cartas, J., Eding, E. H., Schrama, J. W., & Verreth, J. A. (2004). Digestibility, faeces recovery, and related carbon, nitrogen and phosphorus balances of five feed ingredients evaluated as fishmeal alternatives in Nile tilapia, Oreochromis niloticus L. Aquaculture Research, 35(14), 1370-1379.
Morales, G. A., Denstadli, V., Collins, S. A., Mydland, L. T., Moyano, F. J., & Øverland, M. (2016). Phytase and sodium diformate supplementation in a plant‐based diet improves protein and mineral utilization in rainbow trout (Oncorhynchus mykiss). Aquaculture nutrition, 22(6), 1301-1311.
Prachom, N., Haga, Y., & Satoh, S. (2013). Impact of dietary high protein distillers dried grains on amino acid utilization, growth response, nutritional health status and waste output in juvenile rainbow trout (Oncorhynchus mykiss). Aquaculture Nutrition, 19, 62-71.
Yun, B., Xue, M., Wang, J., Sheng, H., Zheng, Y., Wu, X., & Li, J. (2014). Fishmeal can be totally replaced by plant protein blend at two protein levels in diets of juvenile Siberian sturgeon, Acipenser baerii Brandt. Aquaculture nutrition, 20(1), 69-78.
Montanhini Neto, R., & Ostrensky, A. (2015). Nutrient load estimation in the waste of Nile tilapia Oreochromis niloticus (L.) reared in cages in tropical climate conditions. Aquaculture research, 46(6), 1309-1322.
Were these types of papers returned and deemed to not meet the conditions for some reason? If not, the effects of likely excluding many laboratory studies from your search terms on your results should be discussed in greater detail. I am sure you don’t want to re-do your analysis, but I do think it would be much stronger if all of these types of studies were considered.
I’m not convinced of the method used to calculate elemental imbalance. If you divide the %N of an animal by the %N of its diet, what if its growth is P-limited? Or energy-limited? You might get a “high imbalance” by your metric, but it may not mean the same thing as a high imbalance for the limiting nutrient. Has this approach been used elsewhere? Are there previously published approaches that could be used as alternatives?
Results
I was quite surprised that you found that the N:P of egestion did not differ from that of excretion. In the few fish examples I am familiar with, egestion N:P was quite a bit lower than excretion N:P. In those cases I think fish may eat (or were fed) a lot of P in forms they cannot digest (bones or certain forms of P in plants). Your dataset seemed to be skewed heavily towards primary consumers, with few fish examples, and I wonder if your results would hold for higher consumers.
Figure 2. As noted above related to this hypothesis, I’m not sure what to take from this analysis.
Tables – There are several instances where other models were within 2 AIC of the “best” model. It would be worth mentioning in the results that some of these models were not as clearly supported as the best as others, specifically in these cases.
Discussion
Overall I found the discussion of the results to make a lot of sense and didn’t have many concerns with the way this section was written. I did note that some of the paragraphs seemed a bit long so you may consider breaking some ideas up more to help with the flow and to get the key points across.
Line 307 – More information on what pseudofeces are and how they may bias the mollusk results is needed to provide context here.
Line 312 – It’s hard to suggest that trophic mode is not important when your dataset contained only primary consumers.
Reviewer 2 Report
Review of Halvorson and Atkinson “Egestion versus excretion: a meta-analysis examining nutrient release rates and ratios across freshwater fauna”
This paper contrast the relative amounts and ratios of nutrients of egesta vs. excreta in freshwater animals. This topic is relevant and interesting to freshwater ecologists, and the findings that animals in different phyla appear to differ in their release of P uncovers a topic for future research. However, some of the main findings add little to our understanding of ecosystem ecology and seem obvious. For example, the finding that animal egestion rates often exceed excretion rates could have been predicted by any parent or pet owner. The finding that mean egesta contains more N or P is interesting but there is little discussion of how N in egesta relative to DIN in excreta or P in egesta relative to DIP in excreta are comparable in their bioavailability to producers and consumers and thus in their roles in ecosystem function. Along those same lines in Figure 1, what are the possible implications of more inorganic P and N in egesta than excreta? I see that figure and guess that inorganic means less available to organisms, but this is not properly discussed.
I have another major concern with the Results. The figure legends on Figs. 2-4 are all incorrectly matched with the figure (e.g. the legend of Fig. 2 appears to belong with Figure 4). Because of this error, I am not confident in what the figures are showing and I am unable to carefully evaluate the results of the review.
My final major concern is that the authors conclude that “at the ecosystem levels, egestion is as equally important as excretion -…” (Line 343). Because of my comment above that there is almost no discussion of the comparability of the form of N or P in excreta vs. egesta or of the relative bioavailability of the different forms of nutrients in the two kinds of animal waste, I don’t think it is valid to state this conclusion. The authors can conclude that the rates of production of the two kinds of waste appear to be comparable and the total N and P content as well, but without knowledge of how the forms of these nutrients in egesta vs. excreta differ and how available they are to other organisms, this conclusion is not tenable.
Editorial and minor concerns:
Line 11: First sentence of the abstract is long and onerous. I suggest breaking into 2 sentences :``In aquatic settings, animals directly impact ecosystem functions through excretion of dissolved nutrients. However the comparative role of egestion as an animal-mediated nutrient flux is less often studied and is often assumed to be less ecologically impactful. “ Also change impactful to something less evocative of an asteroid (perhaps important).
Line 21: Likewise, last sentence of the abstract is long and difficult to follow. I suggest “ Given egestion is dynamic and can dominate animal-mediated nutrient fluxes, our synthesis demonstrates the need for additional studies of rates, stoichiometry and impacts of animal egestion. (omitted last 10 words)
Line 31, replace “at which animals release nutrient wastes” with “of nutrient wastes by animals”
Line 34: N & P contents of what?
Line 41: you refer to organic nutrient wastes, but both excreta and egesta are organic nutrient wastes, replace with egesta?
Line 42 “These…” is ambiguous, replace with “Egested..”
Line 44: ditto comment for Line 41 above, perhaps use “feces” instead of “waste”.
Line 53: too wordy: edit to ”Animal egestion contrasts with animal excretion…”
Last paragraph of Introduction: I find the hypotheses contrived and arbitrary because no justification for the hypotheses (really predictions) is given. I urge the authors to restate each hypothesis/prediction as a question to improve clarity and omit potential biases. For example, restate hypothesis 1 as the question: “ What is the relative contribution of excretion vs. egestion rates across animal taxa?”
Not only are figure legends on Figs. 2-4 are all incorrectly matched with the figures, but there are other problems with the figures too. For all Figure legend and Table captions, I strongly urge the authors to re-write the legends to give the punchline of what each figure is showing. This approach, advocated by Kroodsma (2000; DE Kroodsma, 2000, A Quick Fix for Figure Legends and Table headings, The Auk 117: 1081-1083) helps the readers to get much more out of viewing figures and tables and helps the authors reinforce their main take-home points.
Figure 1 and 3: X axis labels “excreted from” must be changed because the figures address egestion and excretion so perhaps change to “released”
Round 2
Reviewer 1 Report
The authors have done a fine job of addressing the concerns I raised. While excluding aquaculture studies does feel like a bit of a missed opportunity, I understand the time constraint in searching through that high volume of articles for relevant papers.
I appreciate the addition of citations to back up the elemental imbalance metric used, although I still find it suspect. Perhaps a caveat could still be included in the methods or discussion that this is only meaningful when the imbalance exists for the limiting nutrient?
Aside from this, I am happy to recommend publication of this manuscript.
Author Response
Thank you. We have revised accordingly, adding a point that this calculation assumes the identified nutrient is limiting and this calculation does not account for stronger limitation by other elements, for example when both N and P are highly imbalanced (lines 145-149).
Reviewer 2 Report
The authors adequately addressed most of my suggestions.
In re-reading I found one sentence that did not make sense (line 308):
Broadly, DIN and DIP in excreta should be more bioavailable than N or P in egesta, because the former is mostly organic and requires breakdown by heterotrophs [12].
Do the authors mean "latter" instead of "former"? why would something that is more bioavailable require breakdown? If this is correctly stated, then it is confusing and seemingly illogical.
Author Response
Thank you. This is correct, and it is a typo in the use of latter vs. former. We have revised to using "latter" here because we meant to communicate that organic material is less bioavailable because it requires breakdown by heterotrophs (lines 284-286).